# Understanding the Mechanisms of Fe Deficiency in the Rhizosphere to Promote Plant Resilience

**DOI:** 10.3390/plants12101945

**Published:** 2023-05-10

**Authors:** Zoltán Molnár, Wogene Solomon, Lamnganbi Mutum, Tibor Janda

**Affiliations:** 1Department of Plant Sciences, Albert Kázmér Faculty of Mosonmagyaróvár, Széchenyi István University, H-9200 Mosonmagyaróvár, Hungary; 2Agricultural Institute, Centre for Agricultural Research, H-2462 Martonvásár, Hungary

**Keywords:** rhizosphere, iron deficiency, iron acquisition, microorganisms, interaction

## Abstract

One of the most significant constraints on agricultural productivity is the low availability of iron (Fe) in soil, which is directly related to biological, physical, and chemical activities in the rhizosphere. The rhizosphere has a high iron requirement due to plant absorption and microorganism density. Plant roots and microbes in the rhizosphere play a significant role in promoting plant iron (Fe) uptake, which impacts plant development and physiology by influencing nutritional, biochemical, and soil components. The concentration of iron accessible to these live organisms in most cultivated soil is quite low due to its solubility being limited by stable oxyhydroxide, hydroxide, and oxides. The dissolution and solubility rates of iron are also significantly affected by soil pH, microbial population, organic matter content, redox processes, and particle size of the soil. In Fe-limiting situations, plants and soil microbes have used active strategies such as acidification, chelation, and reduction, which have an important role to play in enhancing soil iron availability to plants. In response to iron deficiency, plant and soil organisms produce organic (carbohydrates, amino acids, organic acids, phytosiderophores, microbial siderophores, and phenolics) and inorganic (protons) chemicals in the rhizosphere to improve the solubility of poorly accessible Fe pools. The investigation of iron-mediated associations among plants and microorganisms influences plant development and health, providing a distinctive prospect to further our understanding of rhizosphere ecology and iron dynamics. This review clarifies current knowledge of the intricate dynamics of iron with the end goal of presenting an overview of the rhizosphere mechanisms that are involved in the uptake of iron by plants and microorganisms.

## 1. Introduction

Iron (Fe), the fourth most abundant and necessary micronutrient for the growth of plants and other organisms, is insoluble in neutral and alkaline soils, making it unavailable to plants. Fe is considered a vital element in the plant system for controlling life-sustaining processes, such as respiration, nitrogen fixation, photosynthesis, assimilation, the synthesis and repair of nucleotides, metal homeostasis, hormonal regulation, and chlorophyll production due to its redox-active nature under biological circumstances [1,2,3,4,5]. Fe can exist in two different oxidation states (Fe^2+^ and Fe^3+^), and it can switch between them by receiving and giving away electrons. In crucial metabolic pathways, such as respiration and photosynthesis, which are needed for plants to make energy, iron plays a key role in enzyme reactions requiring electron transfer [2,6].

Although total Fe is a highly plentiful element in the soil, its accessibility to plants is generally quite low [7]. Thus, insufficiency of iron is one of the most significant limiting variables that influences crop yields, the quality of food, and human nutrition. Inadequate iron absorption results in interveinal chlorosis, stunted growth, reduced nutritional value, and diminished plant yield. Iron deficiency-induced anemia, one of the world’s most common nutritional disorders, requires adequate iron levels in food crops [8]. According to reports, one third of the world’s farmed lands suffer from Fe deficiency, resulting in a considerable annual drop in agricultural productivity, especially in calcareous soils [9,10].

Fe deficiency causes major changes in a plant’s physiology and metabolism, slowing plant growth, impacting nutritional quality, and lowering yield [11], which, in turn, affect the health of people through the food chain, especially those with diets high in plant-based foods.

Plants must boost soil iron’s mobility to overcome its limited availability. In response to Fe deficiency, plants have evolved sophisticated systems to maintain cellular Fe homeostasis by modifying their physiology, morphology, metabolism, and gene expression to enhance Fe availability [12,13]. Basically, these methods involve: (i) acidification, which is facilitated by the secretion of organic acids or protons; (ii) chelation of Fe^3+^ by ligands, which may include siderophores that have an extremely strong affinity for Fe^3+^; and (iii) the reduction of Fe^3+^ to Fe^2+^ through the action of reductases and reducing substances [14,15,16].

Most of the physical, chemical, and biological activities in soil are linked to the geochemistry of iron (Fe) [17] and, consequently, to how much iron is available to the microorganisms and plants that grow in the soil. Iron is mostly found in the rhizosphere as Fe^3+^, which is inaccessible to plants. The rhizosphere is a thin, dynamic zone with substantial abiotic and biotic interactions between soil microorganisms and plant roots [18]. Plant metabolism strongly influences the rhizospheric environment through the release of 5–21% of photosynthetic material by root exudates [19]. Rhizosphere activities and the rhizosphere’s impact on plants are mostly controlled by the release of a complex combination of low and high molecular weight compounds from roots, such as carbohydrates, amino acids, organic acids, protons, phytosiderophores, enzymes, and phenolics [20]. Several microorganisms that interact with plants release siderophores in response to an iron deficiency [21]. These substances can modify the physical, biological, and chemical properties of the soil near the roots.

To improve the adaptability of economically significant plants to specific environmental and soil chemical conditions, it is necessary to comprehend the processes that govern the expression of Fe-deficiency responses in plants. Fe insufficiency is a common issue for various crops, especially those grown in calcareous soils, and it is one of the most significant factors reducing crop production. Increasing our knowledge on how plants deal with iron stress will help us to produce more stress-resistant crops in the future. This article discusses the potential for plant roots and microbes to be able to help plants absorb more iron in iron-deficient soils. The significant role of microorganisms in plant iron uptake has been highlighted by growing knowledge of the interaction between microbes and plants related to dynamics of iron in the rhizosphere.

## 2. Dynamics of Iron in the Rhizosphere

The dynamics of iron in the rhizosphere are controlled by a combination of factors, including the impacts of soil qualities, the absorption and activities of plants and microorganisms, as well as the interactions between these factors. There are two distinct ways that plants can obtain Fe under iron deficiency, namely strategy I and strategy II, to obtain iron from the rhizosphere in an efficient manner [22,23]. In strategy I, Fe is mobilized by the reduction system in most non-graminaceous species. The initial phase of this technique is to acidify the rhizosphere through the H^+^ translocating P-type ATPase AHA2 [12]; this acidification increases Fe^3+^ solubility. To deal with the effects of Fe-deficiency stress in strategy I, plants often induce ferric chelate reductase and Fe(II) transporter in their root systems, acidify the rhizosphere media, and exude organic substances such as phenolics [24,25].

Strategy II (grass species) extracts iron from the soil via a chelation technique. This mechanism is highly reliant on the release of phytosiderophores (PSs) (such as mugineic acids, avenic acid, nicotinamine, etc.) by the root, which would result in the formation of stable iron-phosphate chelates. The Fe(III)–phytosiderophore complex transporter known as yellow stripe1 is a membrane protein that facilitates iron absorption [26]. Temperature, rhizosphere pH, and the type of electrolyte may have a major impact on strategy II Fe acquisition by changing the timing and concentration of the “window of Fe absorption.” [27]. The two strategies rely on molecular mechanisms that are carefully controlled and involve two main components: the root, especially its cell of plasma membrane, and the rhizosphere, which is the soil in closest proximity to the roots [28,29]. Plants have special proteins known as “transporters” that are present in the plasma membranes, and they assist plants in transferring molecules either into or out of the roots as required. Under conditions of severe Fe shortage, phytosiderophores may account for 50–90% of the exudates secreted at the root tip (Fan et al., 1997, as cited in [30]). It is challenging to differentiate between plant species based on their iron uptake mechanism because certain plants, for example, rice or peanut, use both strategies [31,32]. The influence of plants and microorganisms on iron dynamics and status in the rhizosphere, especially on iron solubilization, has been investigated and presented.

### 2.1. Status of Fe in the Rhizosphere and Soil

The rhizosphere is the active zone across a plant root that is home to a diverse population of microorganisms and is impacted by the chemicals produced by plant roots. Rhizosphere processes are the communications between plant roots–soil–microbes that occur and alter continually, impacting things such as nutrient solubility, their movement through the soil, and plant absorption. These systems’ primary driving force seems to be tied to processes of root exudation (Figure 1). Root exudates are organic and inorganic chemicals released by plant roots. They include high and low molecular weight substances, such as carbohydrates, proteins, amino acids, organic acids, protons, polypeptides, enzymes, and hormones, in the rhizospheric soil environment [33,34]. Rhizosphere priming effect occurs when plant roots release recently formed photosynthates into the rhizosphere, which speeds up the breakdown of organic materials by saprotrophic soil bacteria and increases plant nutrient availability [35]. Increased root exudates in the soil improve microbial biomass and soil fertility levels. The dynamics of Fe in the rhizosphere can also be affected by organic compounds generated by the degradation of soil organic matter. These soil microorganisms are essential for the nutrient transformation in the soil and crop plant nutrition absorption. Plants may affect soil qualities by modifying the composition of root exudates, allowing them to adapt and survive under severe environments.

Iron is one of the most plentiful elements in the soil, but after it is weathered, Fe(III) and Fe(II) ions can be released through dissolution and oxidation/reduction. However, when hydroxyl ions (OH^-^) are present, it almost always forms Fe hydroxides and oxides, which have very poor solubility [14]. The dissolution and solubility rates of pedogenic iron oxides (oxides, oxyhydroxides, and hydroxides) play an important role in regulating iron accessibility. The dissolution and solubility rates of iron soil oxides are also significantly affected by pH, microbial population, organic matter content, redox processes, and particle size of the soil [7,14,15,36,37]. Soil pH is the most important of these parameters since it can decrease Fe availability by as much as 95% for every unit increase in soil pH above neutral [38]. When the pH is lowered, the ferric iron is released from its bond with the oxide, making it easier for the roots of plants to absorb it [25]. Fe is transformed to an insoluble Fe–hydroxyl compound in salty, calcareous, alkaline, and sodic soils, which prevents the element from being taken up by plant roots [36]. Soil organic matter level and its breakdown rate affect Fe accessibility because of the formation of excess bicarbonates and phosphates, which hinder the uptake of Fe [4,39].

One of the most significant constraints on agricultural productivity is the low accessibility of iron (Fe) in the soil, which is directly linked to the biological, physical, and chemical activities taking place in the rhizosphere due to the interactions between the soil, microorganisms, and plants [20]. It is widely known that plant roots can alter the pH of the rhizosphere by releasing protons through the H-ATPase enzyme in epidermal cells [40]. This can also occur during Fe deficiency; thus, the plant’s impact on pH can result in the exudation of inorganic metals through the plant roots. Iron deficiency causes soil organisms to emit organic (carbohydrates, amino acids, organic acids, phytosiderophores, phenolics, siderophores, and enzymes) and inorganic (protons) chemicals to improve the solubility of inaccessible Fe pools [20]. Soil pH can be lowered by the plant’s secretion of low molecular weight organic acids [40]. Thus, in order for microorganisms to survive and flourish, the rhizosphere is a geographically and temporally uneven habitat with quick changes in potentially harsh conditions, such as cycles of water stress and anaerobiosis.

It has been extensively documented that metal complexation by humic substances derived from various sources improves plant iron nutrition. The chelation of Fe^3+^ by the organic ligands that comprise the dissolved organic matter has a substantial effect on the solubility of soil iron as well. Reported by [41], more than 95% of the Fe in soil solution is probably complexed or chelated. Depending on the molecular size of humic substances (HS) and solubility, the presence of humified fractions of organic matter in soil sediments and solutions might help provide a reservoir of Fe for plants that exhale metal ligands and supply Fe–HS complexes that are directly utilizable by plant Fe absorption processes [42]. In addition to having iron-chelating qualities, which help enhance iron bioavailability, humic substances also exhibit redox-reactive properties [43].

### 2.2. Iron Interaction with Plant and Rhizospheric Microorganisms

In the rhizosphere, iron competition is important for microbial and plant–microbe interactions. Competition for Fe occurs among microbes and plants, regarding which has the competitive edge due to their capacity to break down plant-derived chelators and their closeness to the surface of the root. However, plants might avoid direct competition with microbes because the amount and type of exudates they release into the rhizosphere change over time and space [30]. Plant to plant interactions, as well as microbial interactions in non-sterile growth circumstances, can modify the iron status of plants. It is well known that the microorganisms in the soil have a substantial impact on the iron nutrition of plants. The iron content of plants can be significantly increased by intercropping grain and legumes [44]. The intercropping of wheat and chickpeas raised the Fe content in wheat seeds [45], whereas the intercropping of maize and peanuts improved the Fe nutrition of peanuts in a calcareous soil [46]. So, we might postulate that the rhizosphere microbes are responsible for the higher iron absorption with intercropped plants.

Rhizosphere microbes live in an environment where plant activity has a substantial effect on the accessibility of nutrients. In the rhizosphere, a wide variety of biotic interactions take place that might influence the composition and diversity of the microbial population in the soil near the roots. These species’ uptake of iron results in complicated interactions, ranging from mutualism to competition [47]. The organization of the microbial community is typically influenced more by biotic interactions in the rhizosphere than by abiotic factors, which are more common in the bulk soil. By excreting rhizodeposits into the rhizosphere, plants provide a fertile and dynamic environment for the microbial populations. The content of iron in solution is further reduced by the iron absorption of these microbes and the host plant. As a result, there is high competition among rhizosphere microbes for iron, encouraging those with the most effective iron absorption strategy [47].

### 2.3. Impact of Plants and Microorganisms on the Iron Status

Plants and microorganisms play important roles in the cycling and availability of iron in the environment. Plant-associated microbes may promote plant development and affect crop output and quality by mobilizing and transporting nutrients [48]. It has been proven that soil microbes play a significant role in promoting plant iron (Fe) absorption in Fe-limiting situations [24,49]. Plant roots and rhizospheric microorganisms release substances such as organic acids, proteins, phenolics, phytosiderophores, and siderophores, which can promote the solubilization of low-availability iron in the soil [14,20,50].

A research report showed that in Chinese cabbage leaves and stalks the concentration of soluble protein, soluble sugar, and vitamin C was significantly decreased under Fe-deficiency stress conditions, whereas the content of cellulose and nitrate was increased [51]. The same study found that Fe-deficiency stress significantly lowered net photosynthetic rate and nitrate reductase activity in the leaves. Iron shortage in the rhizosphere resulted in a 40% rise in root biomass as well as elevated levels of citrate, malate, and phenols in root exudates [52]. The increase in root biomass and elevated levels of these compounds in the root exudates in response to iron deficiency are part of the plant’s adaptive response to this micronutrient limitation.

In Fe-limiting situations, plants and soil microorganisms have used active strategies to enhance soil iron availability, which plays a key role in promoting iron absorption. In the rhizosphere, iron oxides are more easily soluble and dissolvable due to processes such as acidification, chelation, and reduction (Figure 1).

#### 2.3.1. Acidification

Plants that experience a shortage of iron may adopt various methods to enhance their absorption of iron. In strategy I plants, such as Arabidopsis, this involves the release of protons by plasma membrane (PM)-localized H^+^-ATPases (AHAs) to increase acidity in the rhizosphere, which aids in the solubilization of Fe^3+^ [53]. These Fe^3+^ complexes are then reduced to Fe^2+^ and taken up by plants. The process of rhizosphere acidification occurs when H^+^-ATPases, which are bound to the plasma membrane, expel protons from the symplastic area into the rhizosphere, which plays a critical role in nutrient acquisition by plants [5,54]. The FER-like iron deficiency-induced transcription factor (FIT) is crucial for the activation of genes related to iron acquisition and uptake in the root cluster. Its expression is induced in response to iron deficiency and plays a key role in regulating the upregulation of these genes [16,55,56]. FIT is upregulated under low-iron conditions and activates the expression of downstream genes that encode transporters and enzymes involved in iron acquisition, such as the H^+^ -ATPases responsible for rhizosphere acidification and enzymes responsible for iron reduction. This mechanism improves the absorption of iron by enhancing the solubility of iron-containing substances in the soil, creating favorable conditions for iron reduction, and generating a proton motive force that aids in the uptake of irons [15,57].

Acidification helps iron become more soluble in the rhizosphere, which is beneficial to plant health. In the rhizosphere, the roots of plants also release organic acids, such as citric acid, malic acid, and oxalic acid, into the soil. These organic acids can chelate cations, including iron (Fe), making them more available to the plant. Acidification occurs as a result of the secretion of protons and organic acids by microorganisms and plants, resulting in proton concentrations in the rhizosphere that are up to 100-fold greater than in bulk soil [14]. Protons are produced through microbial processes such as nitrification [58]. Plants and microbes release protons into the rhizosphere, which makes it more acidic [59,60].

An insufficiency of iron causes a variety of reactions in soil organisms, releasing inorganic (protons) and organic (carbohydrates, amino acids, organic acids, siderophores, phytosiderophores, enzymes, and phenolics) substances that enhance the solubility of low-accessibility Fe pools. Nevertheless, due to the high microbial activity at the soil–root interface, rhizospheric organic compounds (ROCs) have short half-lives, which may restrict their impact on Fe mobility and acquisition [20]. Overall, the acidification of the rhizosphere caused by proton and organic acid release from plant roots and microbes can increase the availability of iron for the plant and promote optimal growth and development.

Despite the advantages it offers, the degree and adaptability of rhizosphere acidification vary significantly among different plant species. In response to iron deficiency, woody and herbaceous plants such as cork oak (*Quercus suber*), M. plum (*Prunus cerasifera E.*), and cucumber (*Cucumis sativus*) can increase proton extrusion to acidify their rhizosphere [57,61,62]. Similarly, when tomato roots are subjected to Fe deficiency, an increased proton extrusion was detected, and Fe-deficient roots exhibit a greater number of proteins that react with monoclonal antibodies targeting a P-type proton-ATPase from maize compared to Fe-sufficient roots [63]. In contrast, peach–almond hybrids (*Prunus amygdalus B. ₓ Prunus persica B.*), *Vaccinium arboreum* (VA), Southern highbush blueberry (SHB), and wild apple (*Malus baccata*) lack the ability to acidify their rhizosphere through proton extrusion in response to iron deficiency [62,64,65]. Moreover, certain species, such as grapevine (*Vitis vinifera*), display diversity in their responses within the same species, for instance, “Cabernet Sauvignon” can acidify its rhizosphere, while “Balta” cannot [66,67].

#### 2.3.2. Chelation

Chelation, which involves a strategy II-based mechanism, is well known in graminaceous plants. Grasses have the ability to dissolve and absorb Fe from the relatively insoluble inorganic Fe(III) by secreting chelators, which are non-protein amino acids such as mugineic acid and avenic acid. The roots release chelators called phytosiderophores (PSs) that bind with Fe(III) in the rhizosphere and create a complex called Fe(III)–PS, which is transported into the plant via specific plasmalemma transporter proteins [13,68,69]. The release of MAs from plants increases considerably under Fe-deficient conditions, and their capacity to withstand such conditions is closely linked to the quantity of MA that they produce and release. This results in the formation of Fe(III)–PS complexes, which can then be absorbed by the plant roots. The Fe(III)–PS complex is transported into the root cells through the transporters known as yellow stripe (YS) to yellow stripe-like (YSL) after binding to them [70]. Some crops, such as sorghum, rice, and maize, produce only low amounts of deoxymugineic acid (DMA), making them highly vulnerable to Fe deficiency [23]. On the other hand, barley secretes various types of MAs, including mugineic acid, 3-hydroxymugineic acid (HMA), and 3-epi-hydroxymugineic acid (epi-HMA), in relatively high amounts, making it more resistant to low Fe availability [71].

Microorganisms secrete low molecular weight biomolecules that function as chelating agents for metal ions. These molecules are called microbial siderophores and have a greater affinity for Fe^3+^ than phtosiderophores such as mugineic acids, as reported in studies [72]. A significant population of siderophore-producing bacteria is typically found in the rhizosphere, and their secretions enhance the movement and accessibility of metal ions, thereby promoting phytoremediation [73]. When it comes to iron, the existence of microbes that produce siderophores within the rhizosphere has been shown to improve morphological responses. The process by which siderophores mobilize Fe is not yet fully understood. *Cupriavidus necator*, a bacterium that degrades pollutants, has been found to produce a siderophore called cuprabactin, which is utilized by the bacteria to overcome Fe deficiency [74]. This study could be used in the future to introduce microbes to the rhizosphere as a way to help plants acquire more iron. In reaction to an iron deficiency, almost all microbes make siderophores, which are chelating agents for iron. Siderophore-mediated iron absorption is important in regulating the capacity of various microbes to colonize plant roots and promotes microbial associations in the rhizosphere [75]. Siderophores, which are produced by graminaceous plants and by microorganisms, are among those that have a role in the active iron-uptake strategies.

When iron deficiency occurs, the roots of plants begin to secrete phytosiderophores, while citrate has been shown to be an important chelator of iron [76,77]. Iron is also made more soluble in the rhizosphere by its chelation with siderophores and organic acids, which remove ferric iron from insoluble forms and make it accessible to plants (Figure 1) [24,78]. Most aerobic microorganisms make small molecules that have a strong attraction to ferric iron. These molecules, called siderophores, help plants obtain the iron nutrient in iron stress conditions. In addition to being varied in size and chemical makeup, microbial siderophores exhibit a high but variable affinity for ferric iron (Fe^3+^) [79,80]. Hence, microbial siderophores improve the solubility of iron in the soil, which can benefit plant growth and development.

The research suggested that red clover’s iron-deficiency stress may modify the structure of siderophore-emitting microorganisms in the rhizosphere, possibly because of the root’s phenolic secretion, which may increase soil Fe solubility and plant Fe nutrition [24]. Additionally, in rhizosphere soil, a higher concentration of phenolics and number of microorganisms that released siderophores were found in the Fe-deficient treatment than in the Fe-sufficient treatment [24]. However, data have suggested that in a Fe-limited situation, the responses caused by Fe deficiency are not enough for plants to avoid Fe deficiency. According to [81], sunflowers and maize grown in sterile soil grew slowly and had less iron in their tissues than plants grown in non-sterile soil. Similar results were seen for red clover (*Trifolium pratense*) and rape (*Brassica napus*), which grew substantially more slowly in sterile soil and absorbed less iron [82,83]. Furthermore, it was discovered that the barley plant Fe nutritional status affected the microbial population in the rhizosphere, and it was hypothesized that this was affected by variations in root exudates [84]. Hence, it would seem that soil microbial activity is important in promoting plant absorption of iron. Even though some new information has been provided, the precise processes behind the positive influence of microbial siderophores on plant nutrition are still unknown [85].

#### 2.3.3. Reduction

Regarding Fe deficiency, many studies have investigated root responses to this stress condition, showing that most dicot and some monocot species increase NADPH-dependent reductase activity and ATPase-driven proton efflux pumps to solubilize inorganic Fe(III) in the rhizosphere, which enhances Fe(III) reduction to Fe(II) [23,86]. The reduction of Fe(III) in ferric-specific chelates refers to the process of converting Fe(III) ions that are bound to chelating agents or ligands into Fe(II) ions. Reduction of Fe(III) in ferric-specific chelates leads to complex breaking and metal ion release, with the resultant Fe^2+^ being the species of Fe absorbed by the roots [87]. The reduction strategy in the plant Fe uptake process is essential for plant growth and development, as Fe is a critical nutrient required for many important biochemical processes. However, the reduction process is also energy-intensive and can be inhibited by various factors, such as high pH, high levels of calcium, or low levels oxygen [7,15]. As a result, plants have evolved various strategies to cope with Fe deficiency, including the production of various Fe chelators and the modulation of ferric reductase activity. Fe(III) reduction is a necessary and indispensable step in the process of iron uptake by strategy I plant species, since these plants are only capable of absorbing iron in the form of Fe(II). In strategy I, Fe(III) reduction is primarily mediated by reductases, the activities of which are enhanced in response to iron deprivation [14]. H^+^ extrusion into the rhizosphere stimulates its reductase activity in the plasma membrane, and in calcareous soils, high levels of HCO_3_^−^ inhibit it. The reduction strategy in the plant Fe uptake process is based on the ability of the plant to release reducing agents that can convert Fe(III) to Fe(II) in the rhizosphere.

The reduction of Fe(III) in the rhizosphere is mediated by two types of compounds secreted by the plant roots: phenolic compounds and organic acids. Phenolic compounds, such as flavins, coumarins, and catechols, are able to reduce Fe(III) to Fe(II) by donating electrons to the Fe(III) ions. Organic acids, such as citrate, malate, and oxalate, also play a role in Fe(III) reduction by chelating Fe(III) ions and facilitating their reduction by donating protons or electrons. When acidified, membrane-bound ferric reductase oxidase (FRO) such as FRO2 reduces Fe^3+^ to Fe^2+^. Arabidopsis FRO2 was initially recognized as the enzyme that reduces ferric iron chelates at the interface of the root surface and rhizosphere, functioning as a ferric chelate reductase [88]. FRO2 is the specific enzyme accountable for the reduction of Fe(III) chelates in the plasma membrane, which occurs in response to iron deficiency in Arabidopsis roots [89]. The expression of FRO genes is not restricted to the root plasma membrane which suggests that FRO genes play a role in reduction-based Fe transport in other plant organs as well [13,90]. Plants with higher FRO expression exhibit resistance to growth under low iron conditions [91]. Once Fe(III) is reduced to Fe(II) in the rhizosphere, it can be taken up by the root cells through a family of integral membrane proteins called iron-regulated transporter (IRT1) [23,92]. IRT1 is a predominant Fe transporter in *Arabidopsis*, belonging to the ZRT IRT-like protein (ZIP) family. Both IRT1 and FRO2 have specific expression patterns in the root epidermis, where they play crucial roles in the uptake of iron from the soil and are essential for plant growth [93]. The expression of IRT1 is quickly increased upon Fe deficiency, which is likely influenced by signals from both the roots and shoots [94]. These proteins are responsible for the transport of Fe(II) across the plasma membrane into the root cells, where it can be further transported to other parts of the plant.

## 3. Conclusions

Even though iron is one of the most abundant metals on Earth, its bioavailability is considerably decreased due to the low solubility of Fe(III) oxyhydroxide particles, which are the most common form in neutral pH to alkaline soil and under oxygenated conditions. Due to the enormous demand for Fe(III) in the rhizosphere and its poor accessibility in soils, there is intense competition among living organisms for this nutrient. Iron deficiency in the rhizosphere can have a significant negative impact on plant growth and productivity. However, plants have evolved several mechanisms to increase their resilience to iron deficiency and maintain their growth and development, even in low-iron environments. The mechanisms that plants have developed are active techniques for acquiring iron that rely on acidification and reduction of Fe(III) (strategy I) as well as the production of phytosiderophores (strategy II), enabling plants to adjust more effectively to environments with low levels of iron. Plants emit a large portion of their photosynthates as rhizodeposits, which increase microbial population and activity. The roots of plants can influence the rhizosphere microbiome, by forming distinct chemical niches in the soil through the production of phytochemicals (i.e., root exudates), which is dependent on several factors, such as soil characteristics, plant genotype, climatic conditions, and plant nutritional status. During iron deficiency, both plant roots and microorganisms in the soil can release various compounds, such as organic acids, phenolics, siderophores, reductants, and enzymes, into the rhizosphere. By releasing these compounds, plants can mobilize and acquire more iron, ultimately promoting growth and survival in iron-limited environments. As a result, better understanding these dynamics is a critical issue for increasing plant iron nutrition and productivity in sustainable agriculture.

## Figures and Tables

**Figure 1 plants-12-01945-f001:**
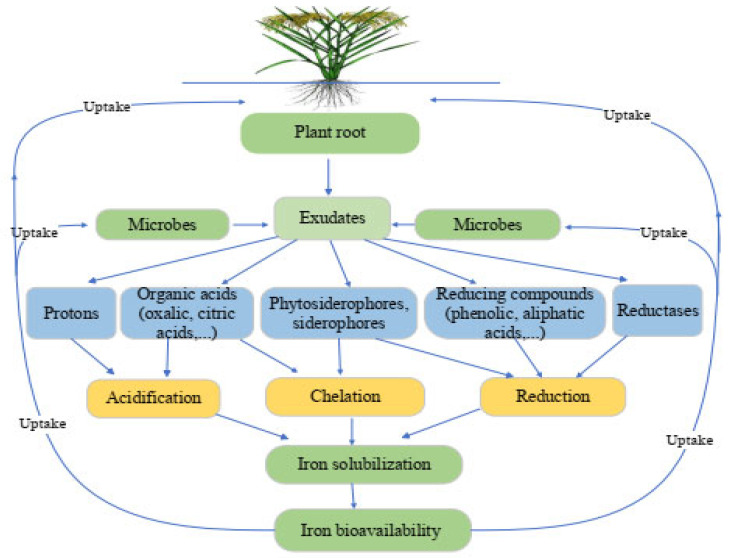
Mechanisms that alter the availability of iron in the rhizosphere. Plants and microbes can improve the bioavailability of iron via (a) acidification—secretion of protons and organic acids, (b) chelation—excretion of complexing molecules with varying affinities for Fe (siderophores, phytosiderophores, carboxylic acids, and phenolics), and (c) reduction—release of substances with reducing characteristics or development of a membrane-bound reductase activity.

## Data Availability

Not applicable.

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
