# Peer review of "Understanding the Mechanisms of Fe Deficiency in the Rhizosphere to Promote Plant Resilience"

_plants, 2023, doi:10.3390/plants12101945_

Round 1

Reviewer 1 Report

The review article "Understanding the Mechanisms of Fe Deficiency in the Rhizosphere to Promote Plant Resilience" is written clear but have severe limitations, For example, The basic idea of the artice is only summerized between the line number 187-250. Overall, it is very concise and limited. I suggest the authors either add more data to this basic themetic area of the article. or make this article as a mini review... srtart form the introduction to the line 187, the data is like an introduction and interactions that only helps to understand Fe role in plants but donot covers the theme of the article. Before publication, i recommend addition of data the above mentioned portion. 

Reviewer 2 Report

This  manuscript presents a concise review of the interactions of plants and their rhizosphere microbiome under Fe limiting conditions and the role of each in enhancing Fe availability to plants. It includes references to several recent research studies as well as other reviews and studies from the past 20 years. I am not sure that the title adequately summarizes the major focus of the manuscript and I would consider modifying it to better summarize the objective stated in the final sentence of the Abstract. It is generally well written with just a few minor corrections of English required (listed below). The references will require some editing to achieve consistency of formatting, in particular in regard to capitalization of journal article titles.

l. 20 which have an important

l. 24 associations

l. 36-40 These two sentences overlap. They could be made more succinct as follows: In crucial metabolic pathways, like photosynthesis and respiration, which are needed for plants to make energy, iron plays a key role in enzyme reactions requiring electron transfer.

l. 55 involve: i) acidification

l. 56 ii) chelation

l. 63 release of 5-21%

l. 77 Dynamics of iron in the rhizosphere

l. 114 secretion of protons and organic acids

l. 118 one of the most plentiful elements in the soil

l. 140 ligands that comprise

l. 180 compounds

l. 190 protons and organic acids

l. 229 and 230 ferric-specific

l. 234 plants

l. 241 there is no need to define rhizosphere here. It was defined earlier in lines 61-62.

l. 261 photosynthates

English is generally excellent with just a few minor revisions required.

Round 2

Reviewer 1 Report

Satisfied with revision

English language is fine